# Natural Deep Eutectic Solvents as a Key Metal Extractant for Fractionation in Speciation Analysis

**DOI:** 10.3390/molecules27031063

**Published:** 2022-02-04

**Authors:** Lena Ruzik, Aleksandra Dyoniziak

**Affiliations:** Chair of Analytical Chemistry, Faculty of Chemistry, Warsaw University of Technology, 00-664 Warsaw, Poland; aleksandra.dyoniziak.stud@pw.edu.pl

**Keywords:** speciation analysis, NADES, young barley, CRM, ICP-MS/MS, GAC

## Abstract

The research aimed to use natural deep eutectic solvents (NADES) as an extractant for fractionation of compounds of selected elements from young barley and to compare it with the fractionation of elements from certified element materials. The use of such a comparison made it possible to prove the possibility of extracting the same forms of elements (species) from different materials, which confirmed the option of using NADES as extractants in speciation analysis. The research was conducted with hyphenated techniques—separation by high-performance chromatography coupled to an isotope-specific detector, mass spectrometry (MS) with ionization in inductively coupled plasma (ICP)—which are widely used in speciation analysis. Natural deep eutectic solvents also help introduce Green Analytical Chemistry principles (GAC). According to the results of our studies, the use of different NADES permit the extraction of various metals from a single sample. Moreover, using other natural solvents of eutectic properties helps extract different species of a given metal.

## 1. Introduction

Speciation refers to various physical and chemical forms in which an element may exist in a real-world material. The forms of an element are its manifestations differing in isotopic composition, electronic structure or oxidation number, or the structure of the molecule or complex [1]. Speciation analysis is the procedure of identifying particular forms of a specific element and their quantification. Correct identification is a challenging task for many reasons. One of them is that it is necessary to determine the properties of a compound of an element present at a trace level, and inadequate sensitivity is the most common drawback of molecule-specific techniques [2]. Another problem is that analyses often seek to determine the identity of compounds that have not yet been described in the literature. The process usually involves several stages, and one of the most important is the specific extraction (leaching) of compounds of the investigated element from the analyzed material.

Besides the capability of effective extraction of chosen analytes, the extractants should: (1) ensure the stability of species of the compounds of interest during extraction and (2) selectively extract well-defined groups of compounds of specified physicochemical properties. Selective extraction permits the characterization of the chemical properties of metal complexes with ligands present in plants and the degree of metal binding in a given tissue. The extraction of chosen compounds using delicate extractants permits investigation of the undecomposed complexes by selection methods in combination with sensitive detection methods [3]. 

In the process of metal extraction from substances of plant origin, extractants of different properties match particular groups of compounds to be extracted [4,5]. The first choice, water, is the most common and fundamental extractant of organic compounds from biological substances. Water can dissolve a large group of substances thanks to its ability to form hydrogen bonds and hydrolysis. The group of substances dissolving in water can be increased using water solutions of appropriate compounds as extractants.

Another class of extractants widely applied for the extraction of compounds present in materials of plant origin are alcohols and their water solutions. The most often used alcohols are methanol [6,7] and ethanol [8,9]. As their polarity is smaller than that of water, they are able to interact with compounds of much more diverse polarities. 

Another extractant used was a water solution of ammonium acetate, a weak acid and weak hydroxide salt used to extract metal complexes with polysaccharides and organic acids and hydrophilic proteins of low molecular mass (close to 17 kDa) [10,11].

An interesting extractant was a water buffer solution of tris(hydroxymethyl)aminomethane (Tris). Tris alters the permeability of biological membranes—the outer coating of cells (the interaction is described on the example of the cells of *Escherichia coli*) [12], which increases the access of extractant to the cell’s inside. The use of Tris permitted the extraction of metal complexes with bioligands present inside the cell and of hydrophilic proteins [13].

In addition, the use of sodium dodecyl sulfate (SDS, SLS) as an extractant was tested in the speciation analysis of various metals. It is a surfactant which interacts with polar and nonpolar solvents thanks to its amphiphilic character. An essential property of SDS as an extractant is the ability to induce local changes in the structure of phospholipid membranes of cells, which permits the release of hydrophobic proteins to the solution [14]. The use of SDS ensures high yields of extraction of macromolecular proteins, higher than the yields obtained with the above-mentioned extractants [15]. The drawback of SDS is the ability to break the noncovalent bonds between the proteins and metal ions. The cleavage does not take place for covalent bonds of metal-proteins, e.g., in selenoproteins [16].

In enzymatic extraction, the extractants were enzymatically active proteins (proteases [17], lipases [18], glycosidases [19] and other hydrolases [20]) that are able to break the chains of proteins, polysaccharides and lipids, which enhances the extraction of analytes from biological matrices. One of the enzymes used was driselase, whose enzymatic activity permitted the release of peptides bonded with the cell wall of noni fruit. In subsequent studies, other enzymes—pectinase and hemicellulose—were applied for the degradation of polysaccharides (pectin and hemicellulose) from the plant cells’ walls [1,21]. Presented results confirmed the possibility of plant cell wall breaking and improved the yield of extraction of copper ions from bilberries [13].

A promising group of extractants proved to be ionic liquids (IL) [22]. They are ionic compounds of melting points lower than 100 °C. Usually, they are built out of large, nonsymmetric organic cations and small organic or inorganic anions. The possibility of using ionic liquids as extractants follows from their ability to solvate many groups of compounds, which is a consequence of the possibility of the formation of different intermolecular interactions: coulomb, dipole–dipole, π–π, n–π, hydrogen bonds, and van der Waals interactions. Prior to the studies of Ruzik et al., there were no reports on the use of ionic liquids to extract metal complexes with bioligands [23].

The successful application of ionic liquids for the extraction of metals and their complexes with bioligands has prompted the search for the possibility of the application of other solvents of eutectic properties [24,25,26]. Natural solvents of eutectic properties (NADES) have become an alternative to organic solvents that are volatile and toxic and easily enter the atmosphere. In comparison to other extractors, NADES show lower toxicity and higher biodegradability. They usually contain two or three substances coming from the groups of sugars, alcohols, organic acids, amino acids and choline derivatives [27,28,29,30]. The molecules of these components are linked by hydrogen bonds which means that the melting point of the obtained natural solvent is lower than the melting points of the substrates. These compounds have been successfully applied in the extraction of compounds from the group of anthocyanins, phenols, flavonoids, isoflavones, DNA, proteins and sugars [31,32,33,34].

NADES have been used for the extraction of metals and their complexes with bioligands. The yield of extraction with these compounds was evaluated on the example of powdered young barley [24]. The signals recorded by size-exclusion chromatography hyphenated with inductively coupled plasma mass spectrometry (SEC-ICP-MS/MS) revealed the extraction abilities of NADES towards a variety of compounds. NADES based on a solution of choline chloride with citric acid, malic acid or glucose can be used for the extraction of metal compounds of intermediate molecular mass, while those based on β-alanine and malic acid show the ability to extract the fraction of compounds of large and small molecular mass. 

In our opinion, the most important achievement in this area of study is providing evidence proving that depending on the extractants applied, dedicated to the extraction of certain complexes, it is possible to perform selective extraction of metal complexes with bioligands. The use of (1) a water buffer solution (Tris) permits extraction of peptides occurring in the cell vacuole and cytosol, (2) the use of enzymes (driselase, pectinase, hemicellulase) permits extraction of peptides and phenolic compounds bonded in the plant cell wall, (3) the use of surfactants (SDS) permits extraction of hydrophobic proteins, (4) the use of ammonium acetate solution permits extraction of organic acids and (5) the use of NADES permits extraction of compounds of low molecular mass, i.e., flavonoids, amino acids or anthraquinone compounds.

The presented investigation uses NADES as an extractant to fractionate compounds of selected elements from young barley and compare it with the fractionation of elements from certified reference material (ryegrass). The use of such a comparison made it possible to prove the possibility of extracting the same forms of elements (species) from different materials, which confirmed the possibility of using NADES as extractants in speciation analysis. The research was conducted with hyphenated techniques—separation by high-performance chromatography coupled to an isotope-specific detector, mass spectrometry (MS) with ionization in inductively coupled plasma (ICP)—which are widely used in speciation analysis.

The SEC-ICP-MS/MS was used to obtain the chromatographic profiles of the extracts whose analysis permitted preliminary fractionalization of metal complexes with bioligands according to the molecular mass of compounds. This procedure allowed—for the first time—the identification of a group of compounds binding given metals in plants [35].

## 2. Results

To investigate the NADES as excellent solvents for the fractionation of various manganese, cadmium, copper, zinc, cobalt and molybdenum compounds, SEC-ICP-MS/MS analysis was performed. The extraction of multiple metal compounds (high molecular weight (HMW) and/or low molecular weight (LMW)) from young barley indicated that, depending on the composition of the NADES, different metals with different efficiency are extracted from the plant (see Figures below). The efficiency of extraction was presented in our earlier studies [24,25].

It is important to mention that investigation about the interaction between the NADES solvents and metals proved that the presence of the extractant was necessary to enhance the metal ion affinity for the hydrophobic phase. Our previous article reviewed this interaction [24,25].

### 2.1. Manganese

Figure 1 compares the manganese isotope content in young barley and ryegrass (Certificate Reference Material—CRM). For the extract using a solvent-based on betaine with citric acid for both young barley and CRM, the highest signals with the retention time at 22 min and 24 min can be observed. It may indicate that manganese in both plants forms complexes with compounds with a 17–44 kDa molecular weight. These compounds can be, for example, proteins or polysaccharides. A signal with a retention time at 38 min was also observed, indicating the presence of manganese connections with LMW, e.g., with organic acids. 

The signal with a retention time at 25 min was observed for the extract with β-alanine and citric acid. It is characteristic for compounds with a 17–44 kDa molecular weight. Additionally, we observed the signal at 38 min. At the same time, there was also a signal observed with a retention time at 22 min for the previous extract and a small signal with a retention time at 17 min, which may correspond to macromolecular compounds with molecular weights 44–158 kDa. 

Similarly, in the extract based on NADES with choline chloride and glucose for young barley, a signal with a retention time at 20 min was observed, characteristic for HMW compounds with a molecular weight of 44–158 kDa. A signal with a retention time at 26 min corresponding to manganese complexes with compounds with a molecular weight of 17–44 kDa and a signal with a retention time at 48 min corresponding to LMW manganese compounds were observed too. On the other hand, for CRM, the extraction efficiency of this solvent is lower, and no single signal can be observed. 

In SEC-ICP-MS/MS chromatograms obtained for the extract using a solvent based on choline chloride with citric acid for young barley, a signal with the retention time at 12 min can be observed. It may indicate the presence of HMW manganese complexes with a molecular weight >670 kDa. The signal with retention time at 22 min corresponds to manganese compounds with a molecular weight of 17–44 kDa. In addition, we observed signals with higher retention times characteristic of small molecule compounds. On the other hand, for CRM, only signals with retention times above 30 min can be observed, indicating that only LMW manganese compounds were extracted by this solvent. 

For the extract with choline chloride and citric acid, both for the young barley sample and CRM, there are signals with a retention time at 22 min and 25 min corresponding to manganese compounds with a molecular weight of 17–44 kDa, as well as signals with higher retention times indicating the presence of LMW manganese compounds.

### 2.2. Cadmium

On the chromatographic profiles for the cadmium isotope (Figure 2) for extracts based on betaine with citric acid, two signals can be found, corresponding to combinations of cadmium with compounds with a molecular weight of 17–44 kDa—a signal with retention time at 22 min. A signal with a retention time at 29 min was also obtained in both cases. It probably indicates the presence of cadmium complexes with compounds with a molecular weight of 1.35–17 kDa. 

We could observe three signals with the same exact retention times as above for the extracts with choline chloride and citric acid. We also observed a signal with a retention time at 42 min, most likely coming from cadmium complexes with LMW compounds. On the other hand, for CRM, we observe a small signal with a retention time at 24 min, characteristic for HMW compounds with a molecular weight of 17–44 kDa, and a signal with a retention time at 29 min, probably indicating the presence of connections of cadmium with compounds with a molecular weight of 1.35–17 kDa. 

On SEC-ICP-MS/MS chromatograms obtained for the extract of young barley with NADES based on β-alanine with citric acid, a small signal with the retention time at 24 min was observed. It indicates cadmium complexes with macromolecular compounds with a 44–158 kDa molecular weight. In addition, we observed two signals with retention times at 22 min and 25 min, which indicated the presence of cadmium compounds with a molecular weight of 1.35–17 kDa. It can be concluded from the chromatographic profiles that the extraction efficiency of these compounds by this solvent is lower than when using NADES with choline chloride with citric acid and betaine with citric acid. For CRM extract, two small signals were observed—the first signal with a retention time at 25 min and the second signal with a retention time at 29 min, indicating the extraction of cadmium compounds with a molecular weight of 17–44 and 1.35–17 kDa, respectively.

In the case of using NADES based on choline chloride with glycerol for young barley, a signal with the retention time at 22 min can be observed, indicating the extraction of cadmium compounds with a molecular weight of 17–44 kDa. While using NADES based on choline chloride with glucose, a signal of a retention time of 48 min may indicate the presence of LMW cadmium compounds. No signals were observed for these two CRM extracts, which may show the low extraction efficiency of cadmium compounds present in this plant material.

### 2.3. Copper

On the chromatographic profiles for the copper isotope (Figure 3) for extracts based on betaine with citric acid, signals were observed with retention times at 21 and 25 min, indicating that copper in both plants forms complexes with compounds with a molecular weight of 17–44 kDa. There is also a signal with a retention time of 29 min, which may indicate the presence of copper connections with compounds with a molecular weight of 1.35–17 kDa. There is also a signal with a retention time at 39 min, which probably indicates the presence of connections of copper with low-molecular compounds. 

In the case of young barley extract with NADES, based on choline chloride with citric acid, signals with a retention time at 22 and 25 min can be noticed. It may indicate the presence of copper compounds with a molecular weight of 17–44 kDa. There is also a signal with a retention time of 30 min, proving the presence of copper compounds with a molecular weight of about 1.35 kDa in the extract. A signal with a retention time of 42 min was also noticed, indicating the presence of copper complexes with LMW compounds. In the case of CRM, a signal with a retention time at 25 min was observed, characteristic for compounds with a molecular weight of 17–44 kDa, and a signal with a retention time at 29 min, which may correspond to copper complexes with compounds with a molecular weight of 1.35–17 kDa, as well as the signal with the retention time at 41 min, which may indicate the presence of LMW copper compounds. 

On SEC-ICP-MS/MS chromatograms obtained for the extract of young barley with NADES based on β-alanine with citric acid, signals with retention times at 21 and 25 min were observed. This may indicate the presence of copper complexes with compounds with a molecular weight of 17–44 kDa. In addition, a signal with a retention time at 37 min indicates the presence of LMW copper compounds. On the other hand, there is also a signal with a retention time of 25 min for CRM extracts. In addition, we observed a signal with a retention time at 38 min, characteristic of low molecular compounds.

In the case of the NADES extract based on choline chloride with glycerol, both for young barley and CRM, a signal with a retention time at 25 min is visible as well as signals with higher retention times, probably indicating the presence of copper connections with LMW compounds.

Additionally, on SEC-ICP-MS/MS chromatograms for young barley with choline chloride with citric acid, a signal with a retention time at 28 min was observed. It indicates the presence of copper complexes with compounds with a molecular weight of 1.35–17 kDa and signals with higher retention times, probably indicating the presence of copper compounds with a molecular weight of 1.35–17 kDa. In the case of the CRM extract, the obtained chromatographic profile has a very low intensity, indicating a low extraction efficiency of copper compounds present in this plant material.

### 2.4. Zinc

On SEC-ICP-MS/MS chromatograms obtained for the zinc isotope (Figure 4) for extracts based on betaine with citric acid, a signals were observed with 21 min and 24 min retention times. It may indicate that zinc in both plants forms complexes with compounds with a 17–44 kDa molecular weight. We could also observe a signal with a retention time at 27 min, indicating the presence of zinc complexes with compounds with a molecular weight of 1.35–17 kDa. In addition, we observed a signal with a retention time at 39 min, which may indicate the presence of LMW zinc compounds. For extracts based on choline chloride with citric acid of both young barley and CRM extracts, a signal with a retention time at 24 min was observed as well as signals with higher retention times, 42 min, indicating the presence of LMW zinc compounds. 

For young barley extracts and CRM extracts with NADES based on β-alanine with citric acid, there were also observed signals with retention times at 24 min and 27 min, appearing for the previous extracts. In addition, signals with higher retention times can be observed—for young barley at 36 min and for CRM at 38 min, which may be evidence of the combination of zinc with LMW compounds. 

Moreover, on SEC-ICP-MS/MS chromatograms for the extract based on NADES with choline chloride with glycerol, a signal with the retention time at 24 min can be observed for both extracts. However, a signal with a retention time at 22 min, characteristic for compounds with a molecular weight of 17–44 kDa, was observed only for young barley. There is also a signal with a retention time at 36 min, which may indicate the presence of LMW zinc compounds. 

In the case of young barley and CRM extracts with NADES based on choline chloride with glucose, the obtained chromatographic profiles have a lower intensity than the other extracts, indicating a low efficiency of zinc compound extraction by this solvent.

### 2.5. Cobalt

In Figure 5 we compared chromatograms for cobalt isotope content in the respective extracts of young barley and CRM. Signals with 21 min and 24 min retention times extracted betaine with citric acid for young barley. This may indicate that cobalt forms complexes with a 17–44 kDa molecular weight. There are also signals with higher retention times that may indicate connections of cobalt with LMW compounds. CRM signals with the exact retention times and a signal with a retention time at 14 min are noticeable, indicating the presence of HMW cobalt complexes with a molecular weight of 158–670 kDa in the extract. 

On SEC-ICP-MS/MS chromatograms a signal with the retention time at 24 min and signals with higher retention times were obtained for the extract of choline chloride with citric acid for both extracts, the intensity of these signals being higher than in the case of other extracts. For extracts based on β-alanine with citric acid, a signal with a retention time at 24 min was also observed, and the signal with the highest retention time at 39 min was observed. 

For extracts of young barley with choline chloride with glycerol, a signal with a retention time at 17 min was observed, indicating the presence of cobalt complexes with HMW with a molecular weight of 44–158 kDa. In addition, a signal with a retention time at 21 min, with low intensity, was observed for the extract of betaine with citric acid. In the case of CRM, no clear signals were observed for this solvent, from which it can be concluded that it extracts cobalt compounds with low efficiency. 

The low extraction efficiency of cobalt compounds can also be observed in the case of NADES based on choline chloride with glucose. The chromatographic profiles for young barley and CRM are very low intensity.

### 2.6. Molybdenum

On SEC-ICP-MS/MS chromatograms (Figure 6) obtained for molybdenum compounds in the extract with NADES based on choline chloride with citric acid, signals with retention time at 23 min and 26 min were observed. This may indicate that molybdenum forms complexes with compounds with a molecular weight of 17–44 kDa, a signal with a retention time at 28 min, which may reveal the presence of connections with compounds with a molecular weight of 1.35–17 kDa. In addition, we observe a signal with a retention time at 38 min, probably indicating the presence of LMW molybdenum compounds. For CRM extracts, a signal with a retention time at 19 min was observed, indicating the presence of HMW molybdenum complexes with a molecular weight of 44–158 kDa. We can also observe signals with retention times equal to those for young barley extract at 23 min, 26 min and 28 min. There is also a signal with 42 min retention time, characteristic of LMW compounds. 

For extracts based on betaine with citric acid, a signal with a retention time at 29 min can be found, indicating the presence of molybdenum connections with compounds with a molecular weight of 1.35–17 kDa as a signal with a retention time at 38 min. Moreover, a signal with a retention time at 42 min was observed for CRM extracts. Both signals may correspond to LMW molybdenum compounds. 

For the extract of young barley with NADES based on β-alanine with citric acid, a signal with the retention time at 28 min was also observed. As well as a signal with a retention time at 37 min, characteristic for LMW compounds. For CRM extracts, signals with retention times were observed, which occurred in the previously described extracts at 23 min, 28 min and 39 min, probably proving the presence of molybdenum compounds with LMW compounds. 

For young barley extract based on choline chloride with glycerol, a signal with a retention time at 24 min can be found, indicating the presence of molybdenum connections with compounds with a molecular weight of 17–44 kDa, as well as a signal with a retention time at 38 min. For the CRM extract, a small signal with the retention time at 23 min was observed and a signal with the retention time at 36 min, indicating the presence of LMW molybdenum compounds in the extract. 

In the case of young barley extract with NADES based on choline chloride with glucose, a signal with a retention time at 29 min, indicating the presence of molybdenum connections with compounds with a molecular weight of 1.35–17 kDa, and a signal with a retention time at 43 min, can be observed, probably due to the extraction of LMW molybdenum compounds by this solvent. For CRM extracts, the obtained chromatographic profile is characterized by low intensity, indicating a low extraction efficiency of molybdenum compounds by this solvent.

## 3. Discussion

The first step of our investigation was to select natural solvents with eutectic properties (NADES) to extract metal compounds from plant tissue. It was important to find the natural solvent that enables efficient metal extraction from plant samples, is environmentally friendly and complies with Green Analytical Chemistry (GAC) principles; they are less toxic than other popular solvents. Our study chose five different NADES solvents and observed the fractionation of six various metal complexes using SEC-ICP-MS/MS techniques.

On SEC-ICP-MS/MS chromatograms of the extracts from young barley and ryegrass (CRM), we observed signals assigned to compounds of intermediate molecular mass. NADES based on choline chloride solution with citric acid or glucose can extract intermediate molecular mass metal compounds. In contrast, those solvents based on β-alanine and citric acid show the ability to extract the fraction of compounds of high and low molecular mass. 

However, on SEC-ICP-MS/MS chromatograms obtained for NADES based on betaine, we could observe the ability to extract metal complexes with small bioligands, e.g., organic acids. On observed SEC-ICP-MS/MS chromatograms obtained for glucose extraction, we could not observe any signals, which indicates the low efficiency of extraction for this NADES solution.

Metals presented in young barley extracts are bound to macromolecular compounds with a 17–44 kDa molecular weight. These compounds were extracted with the highest efficiency by NADES based on choline chloride with citric acid and betaine with citric acid. These compounds can be proteins or polysaccharides. There are also visible signals with higher retention times indicating connections with medium and low molecular weight compounds (e.g., flavonoids or organic acids). Depending on the registered isotope, these signals have different intensities. The isotopes of manganese, cobalt, and molybdenum exhibit the most incredible intensity compared to the other signals, which may indicate that they occur to a greater extent in the form of LMW compounds in extracts than other isotopes. In addition, the chromatograms for most of the studied isotopes show signals of low intensity for low retention times, which may indicate that they form complexes to a lesser extent with compounds with molecular weight above 44 kDa.

The chromatographic profiles of manganese and zinc were characterized by the highest intensity—according to the literature data, the content of these metals in the tested plant (young barley and ryegrass) is the highest. We observed the same tested isotope with the exact retention times for the same tested isotope, indicating that using NADES, regardless of the material tested, the same groups of compounds are extracted.

According to the results of our studies, the use of different NADES permits the extraction of various metals from a plant sample. Moreover, using other natural solvents of eutectic properties helps extract different species of a given metal. This possibility allows the planning of extraction of a sequence of metals and their compounds from plants. It enables speciation analysis of compounds bounded to insoluble parts of plants. Moreover, the application of sequential extraction with NADES permits effective extraction of metals to perform speciation analysis of biological and medical materials. 

The wide use of NADES as extractants for many groups of compounds has now been extended with the possibility of using them as extractants for metals in the speciation analysis of selected metals in natural materials.

It is important to mention that NADES helps to introduce a few of the 12 principles of GAC [36]: (1) The NADES reagents are obtained from renewable sources, and, thanks to this, (2) toxic reagents can be eliminated and replaced; also use of the presented hyphenated techniques helps in other respects: (3) SEC-ICP-MS/MS techniques as a multi-analyte technique are preferred versus techniques which can determine only one analyte at a time and (4) during the presented analysis a minimal sample and a minimal number of samples were used.

## 4. Materials and Methods

*Chemicals and materials:* The dried, ground, young barley was purchased from Intenson (Poland). Choline chloride (≥99.98%), glycerol (≥99.5%), betaine (≥99%), citric acid (≥99.5%), glucose (≥99.5%), β-alanine (≥99%), ammonium acetate (≥99.995%) and acetic acid (≥99.98%) of analytical reagent grade were purchased from Sigma Aldrich (USA). CRM ERM-CD281 Rye Grass was purchased from Institute for Reference Materials and Measurements (IRMM), Belgium. High purity water (15 MΩ cm) was obtained with Milli-Q Elix 3 Water Purification system Millipore (France). The SEC column was calibrated using the size exclusion standard (BIO-RAD, Poland). The calibration curves were prepared using a solution of Environmental Spike Mix (1000 mg L^−1^ of Fe, K, Ca, Na, Mg and 100 mg L^−1^ of Ag, Al, As, Ba, Be, Cd, Co, Cr, Cu, Mn, Mo, Ni, Pb, Sb, Se, Tl, V, Zn, U; matrix 5% HNO_3_) purchased from Agilent Technologies (USA).

*Preparation of NADES:* The following procedure of NADES synthesis is based on the heating method [37] to obtain natural deep eutectic solvents (NADES) with a volume of ~10 mL. The presented method was employed to obtain NADES with a known amount of water. The two-component mixture with a calculated water amount was placed in a bottle with a stirring bar and cap and heated in a water bath below 50 °C with agitation until a clear liquid was formed (about 30–90 min). The molecular formula of used HBA and HBD for the formulation of NADES is presented in Figure 7.

*Instrumentation:* Chromatographic separations were performed using Agilent 1100 gradient HPLC pump (Agilent Technologies, Waldbronn, Germany). All connections were made of PEEK tubing (0.17 mm i.d.). As an element-specific detector for the online HPLC detector, Agilent 8900 ICP Triple Quadrupole Mass Spectrometer (Tokyo, Japan) was used throughout. The spectrometer, with the Pt-cones in the interface, was equipped. The torch position and nebulizer gas flow were adjusted daily, emphasizing the decrease of the level of CeO^+^ (below 0.2%) to minimize the risk of occurrence of the polyatomic interferences caused by oxides. The instrument was equipped with Pt sampling and skimmer cones, a MicroMist nebulizer, Scott spray chamber, and quartz torch with 2.5 mm i.d. injector. Samples were introduced directly into the ICP-MS with the standard peristaltic pump. Analyses were performed in Time-Resolved Analysis (fast TRA) mode, using a dwell time of 0.1 ms (100 μs) per point with no settling time between measurements. The RF power was 1550 W, nebulizer gas flow—1.03 L min^−1^, collision gas flow (helium)—5.0 mL min^−1^, sample uptake rate 0.35 mL min^−1^, sample depth 8.0 mm, and monitored isotopes: ^55^Mn, ^111^Cd, ^63^Cu, ^66^Zn, ^59^Co, ^95^Mo. The working conditions were optimized daily using a 1 µg L^−1^ Li, Y and Tl solution in 2% (*v*/*v*) HNO_3_. 

The fractionation for the metal species was performed using size exclusion chromatography coupled to ICP-MS/MS. Metal species were eluted from SEC Superdex200 10/300GL (GE Healthcare Life Sciences, Freiburg, Germany) column with 10 mM ammonium acetate buffer (pH 7.4) as a mobile phase, isocratic mode with −7 mL min^−1^ flow. Before the analysis the column was calibrated with a mixture of thyroglobulin (670 kDa), γ-globulin (158 kDa), ovalbumin (44 kDa), myoglobin (17 kDa) and vitamin B12 (1.35 kDa). 

The validation parameters, such as precision and accuracy, were assessed. The method’s precision was evaluated by analyzing ten independent experimental preparations for each metal—test sample made against the internal standard and the %RSD of metals calculated. The accuracy of the obtained data was high and repeatable (%RSD) and was in the range of 0.74 to 3.41%.

A Bandelin Sonorex Model 1210 ultrasonic bath (Bandelin, Berlin, Germany), MPW Model 350R centrifuge (MPW Warsaw, Poland), water bath with thermostatically controlled temperature (Mammert, Germany) and sonication probe (Bandelin Sonoplus, Berlin, Germany) were used for extraction procedures.

*Extraction procedure:* Grounded samples (0.05 g of dry young barley powder and CRM) were extracted by vortexing (the extraction parameters—Table 1) with 2 mL of each solvent. The obtained solutions were centrifuged for 15 min at 15,000 rpm at 25 °C. The final supernatant was filtered with a 0.45 µm syringe filter (Sigma-Aldrich, Bellefonte, PA, USA). The remaining part of the filtrate was injected into the size exclusion column. 

## Figures and Tables

**Figure 1 molecules-27-01063-f001:**
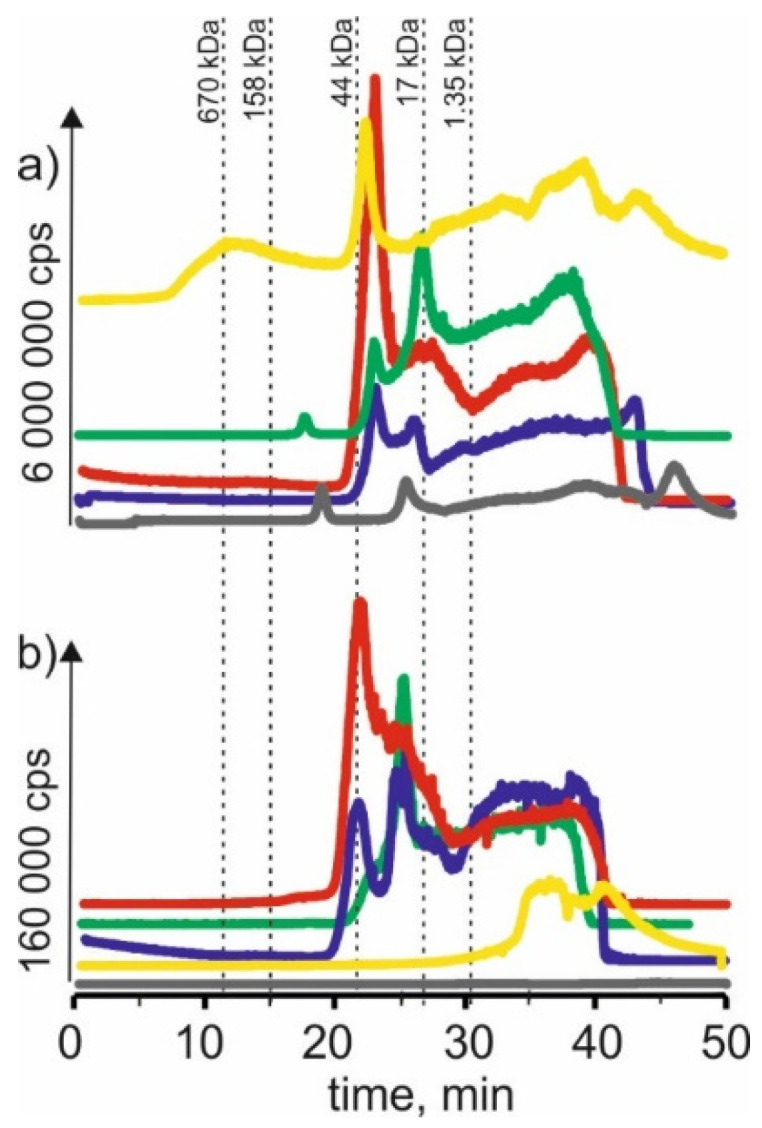
SEC-ICP-MS/MS chromatograms for manganese, obtained for extracts of (**a**) young barley and (**b**) CRM with different NADES solutions. Registered isotope: ^55^Mn; choline chloride with citric acid (blue line), betaine with citric acid (red line); β-alanine with citric acid (green line); choline chloride with glycerol (yellow line); choline chloride with glucose (grey line).

**Figure 2 molecules-27-01063-f002:**
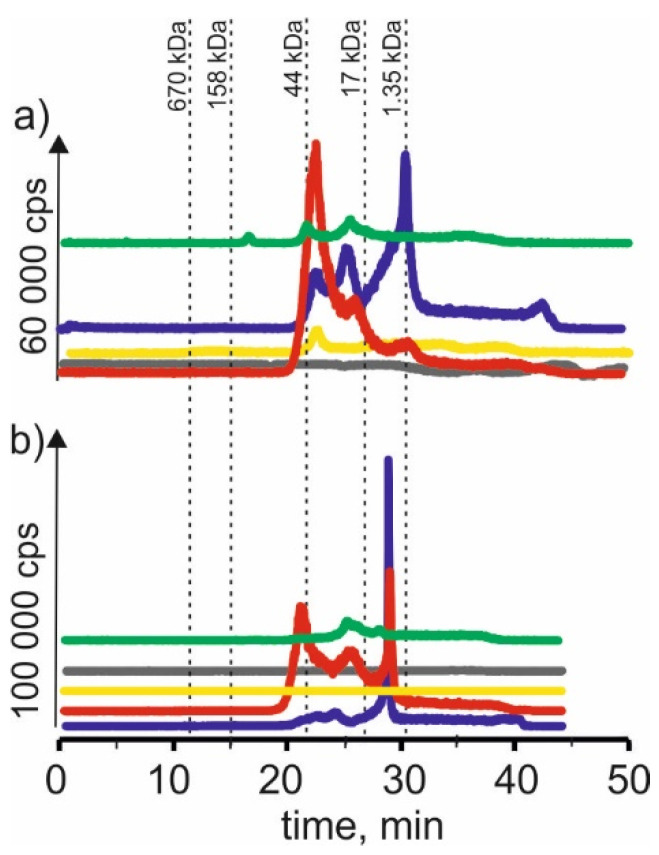
SEC-ICP-MS/MS chromatograms for cadmium, obtained for extracts of (**a**) young barley and (**b**) CRM with different NADES solutions. Registered isotope: ^111^Cd; choline chloride with citric acid (blue line), betaine with citric acid (red line); β-alanine with citric acid (green line); choline chloride with glycerol (yellow line); choline chloride with glucose (grey line).

**Figure 3 molecules-27-01063-f003:**
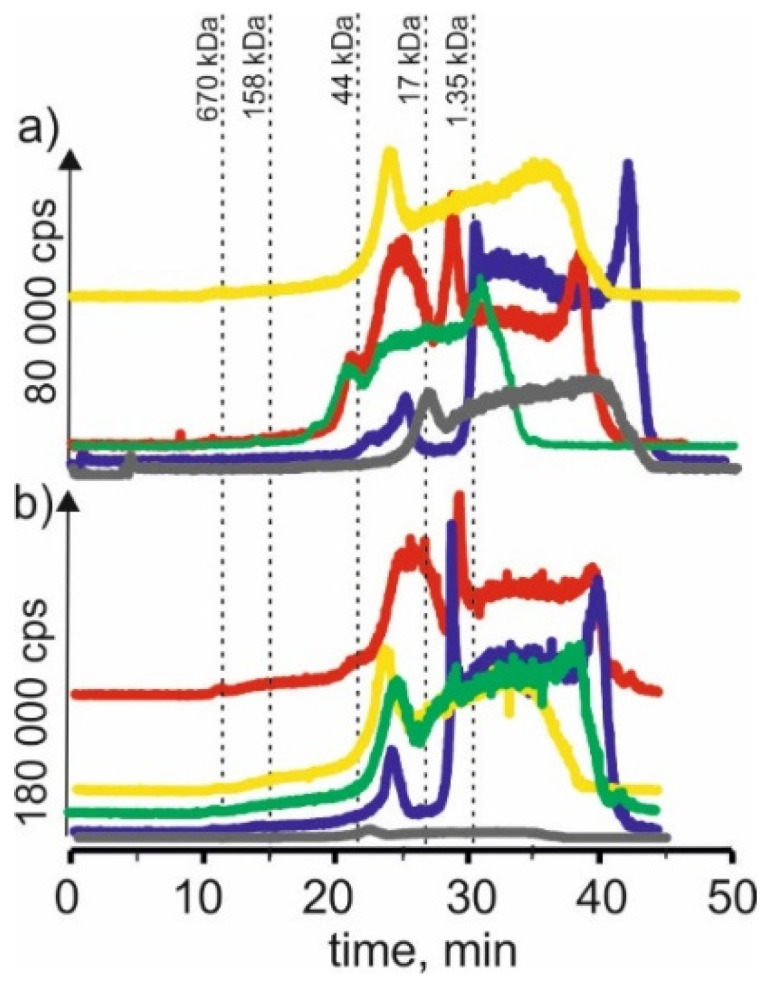
SEC-ICP-MS/MS chromatograms for copper, obtained for extracts of (**a**) young barley and (**b**) CRM with different NADES solutions. Registered isotope: ^63^Cu; choline chloride with citric acid (blue line), betaine with citric acid (red line); β-alanine with citric acid (green line); choline chloride with glycerol (yellow line); choline chloride with glucose (grey line).

**Figure 4 molecules-27-01063-f004:**
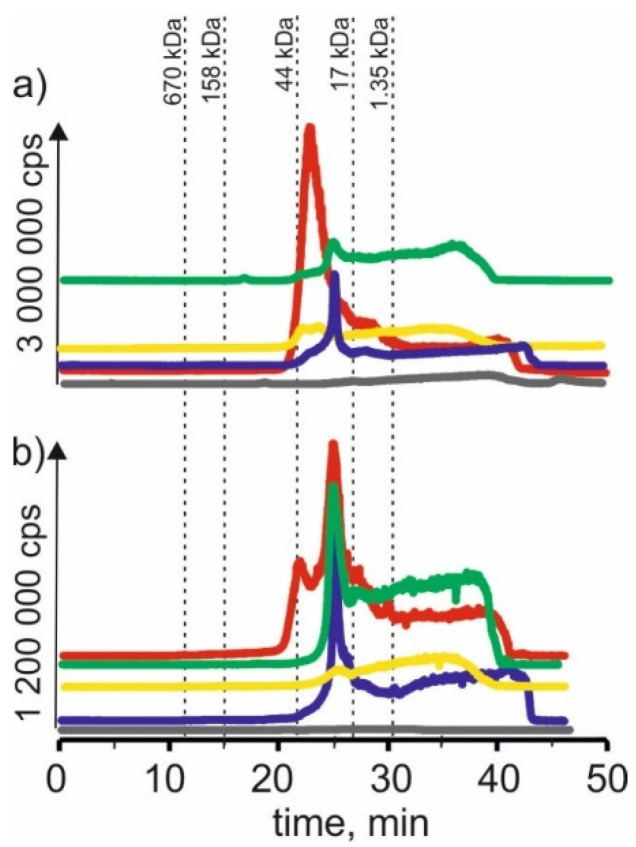
SEC-ICP-MS/MS chromatograms for zinc, obtained for extracts of (**a**) young barley and (**b**) CRM with different NADES solutions. Registered isotope: ^66^Zn; choline chloride with citric acid (blue line), betaine with citric acid (red line); β-alanine with citric acid (green line); choline chloride with glycerol (yellow line); choline chloride with glucose (grey line).

**Figure 5 molecules-27-01063-f005:**
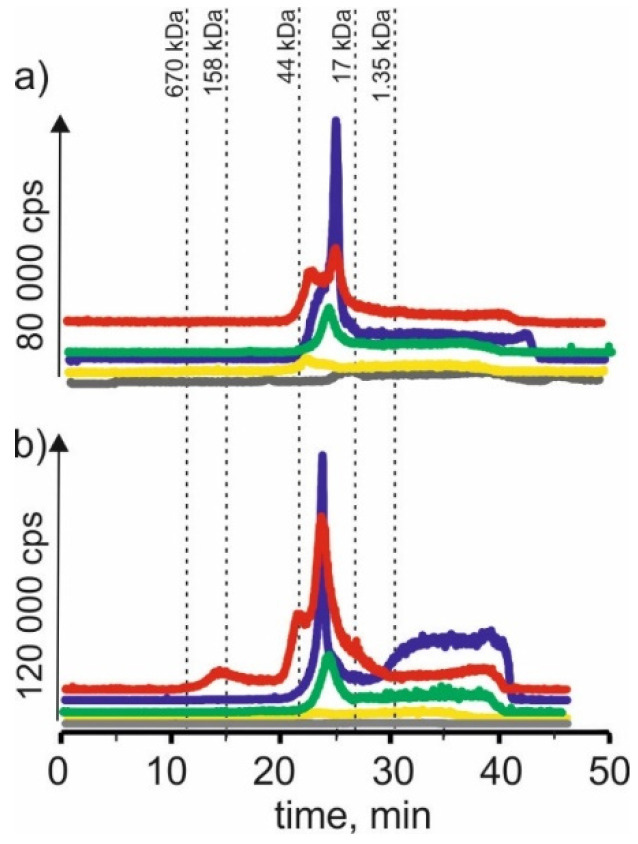
SEC-ICP-MS/MS chromatograms for cobalt, obtained for extracts of (**a**) young barley and (**b**) CRM with different NADES solutions. Registered isotope: ^59^Co; choline chloride with citric acid (blue line), betaine with citric acid (red line); β-alanine with citric acid (green line); choline chloride with glycerol (yellow line); choline chloride with glucose (grey line).

**Figure 6 molecules-27-01063-f006:**
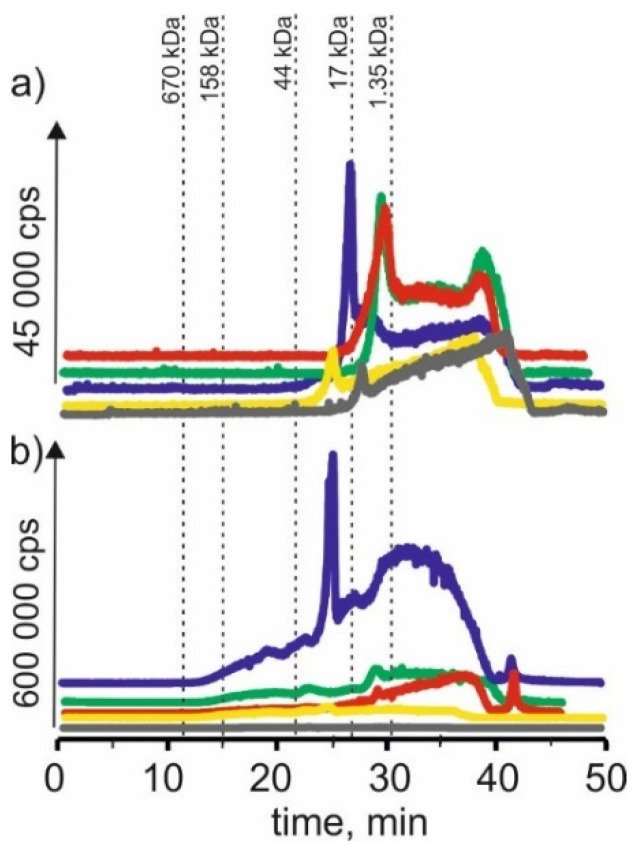
SEC-ICP-MS/MS chromatograms for molybdenum, obtained for extracts of (**a**) young barley and (**b**) CRM with different NADES solutions. Registered isotope: ^95^Mo, choline chloride with citric acid (blue line), betaine with citric acid (red line); β-alanine with citric acid (green line); choline chloride with glycerol (yellow line); choline chloride with glucose (grey line).

**Figure 7 molecules-27-01063-f007:**
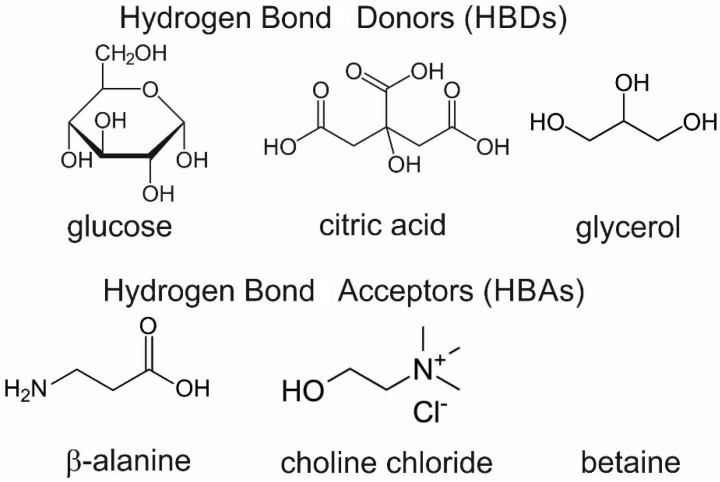
The molecular formula of HBAs and HBDs used for the formulation of NADES.

**Table 1 molecules-27-01063-t001:** The extraction temperature and time of vortexing.

Hydrogen Bond Acceptors	Hydrogen Bond Donors	Molar Ratio	Temperature [°C]	Time [min]
β-alanine	citric acid	1:1	30	40
choline chloride	citric acid	1:1	40	35
betaine	citric acid	1:1	40	35
choline chloride	glucose	1:2	50	40
choline chloride	glycerol	2:1	60	30

## Data Availability

Not applicable.

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
