# Peer review of "Natural Deep Eutectic Solvents as a Key Metal Extractant for Fractionation in Speciation Analysis"

_molecules, 2022, doi:10.3390/molecules27031063_

Round 1

Reviewer 1 Report

This manuscript reports the use of natural deep eutectic solvents (NADES) as the excellent solvent for the fractionation of various manganese, cadmium, copper, zinc, cobalt, and molybdenum compounds. This paper is organized well. Therefore, I suggest that the manuscript can be considered after minor revisions. There are some issues for the attention of the authors.

  1. The molecular formula and configuration of natural deep eutectic solvents used in this paper should be provided with the Figures form.
  2. Some formats should be checked, such as “100°C”.

Author Response

  1. The molecular formula and configuration of natural deep eutectic solvents used in this paper should be provided with the Figures form.

It has been added.

  1. Some formats should be checked, such as “100°C”.

It was checked and rewritten.

Reviewer 2 Report

This article investigated the fractionation of compounds of selected elements from young barley by using natural deep eutectic solvents (NADES). The authors were designed the work systematic way with performing some valuable experimental works. The results were interesting and valuable to the authors of Molecules. However, it should be clarified more clearly in the following questions: 

(1). For topics, avoid abbreviations and formulae where possible. Only abbreviations firmly established in the field may be eligible. Obviously, it is inappropriate to directly use the abbreviation of extractant. Abbreviations in paper may cause confusion for readers.

(2). Although an extensive introduction is provided, authors fail to stress the main points. There is some unnecessary information in the paper which needs to be reduced. Authors should arrive at the solution by a simple elaborations.  

(3). As this is an approach for young barley, it is likely for certain limitations to be encountered. Do the authors have any of such limitations to report? If yes, it should be reported and a suitable approach to remove this limitation can be recommended for future studies in the conclusion.

(4). The paper presents good results on the extraction of different species of a given metal. But very little and poor explanations on the mechanism of extraction process. For example, as the author mentioned in the introduction, the interaction forms between extractant and metal. The reaction mechanism between elements and extractants can give a simple explanation.

(5). The characterization methods may get a bit monotonous, but there'll be enough of it. It is suggested to add the prospect of characterization methods and the applications of natural deep eutectic solvents (NADES) for other substances.

In conclusion, the paper is innovated and interesting, it could be published with minor changes.

Author Response

(1). For topics, avoid abbreviations and formulae where possible. Only abbreviations firmly established in the field may be eligible. Obviously, it is inappropriate to directly use the abbreviation of extractant. Abbreviations in the paper may cause confusion for readers.

We agree with the remark, and the abbreviation from the topis has been changed.

(2). Although an extensive introduction is provided, the authors fail to stress the main points. There is some unnecessary information in the paper that needs to be reduced. Authors should arrive at the solution by a simple elaborations. 

Thank you very much for this remark. We try to find the information we could cross out, but all the presented information is crucial to understanding our investigation and conclusions. We hope that the lack of changes in this paragraph will not change the opinion on the importance of the presented research.

The authors would like to collect a description of the solvents used in the speciation analysis, indicating their advantages and disadvantages and the need to look for new solvents such as the presented NADES. The extraction process in the speciation analysis is a very important, difficult stage of the analytical procedure and the aim of the article was to present NADES as a new solution in the extraction in speciation analysis, therefore it seems to me that the description of other extractants used is important and just indicates the main purpose of the presented research.

(3). As this is an approach for young barley, it is likely for certain limitations to be encountered. Do the authors have any of such limitations to report? If yes, it should be reported and a suitable approach to remove this limitation can be recommended for future studies in the conclusion.

As in the case of plant tissues, we have not encountered any limitations in the case of young barley. The application of NADES was very easy and without a problem, it was possible to extract the metals of interest. Therefore, no restrictions on the use of NADES have been ignored, as we have not encountered them.

 (4). The paper presents good results on the extraction of different species of a given metal. But very little and poor explanations on the mechanism of extraction process. For example, as the author mentioned in the introduction, the interaction forms between extractant and metal. The reaction mechanism between elements and extractants can give a simple explanation.

The information about the interaction between extractants and metals has been added to the paragraph Results.

(5). The characterization methods may get a bit monotonous, but there'll be enough of it. It is suggested to add the prospect of characterization methods and the applications of natural deep eutectic solvents (NADES) for other substances.

It has been added.

Reviewer 3 Report

The reviewed manuscript entiteld “ NADES as a key metals extractants for fractionation in speciation analysis” by Ruzik and Dyoniziak is a valuable attempt of validation the titled procedure for speciation analysis of selected metals from young barley.

Although I do not feel competent for English language correction I feel that paper must be rewritten. The whole introduction is unclear, sentences are too complicated and vague. Please consider rewriting for increase clarity and enabling better comprehension. The acronym SEC-ICP-MS is first time used in line 102 but explained in line 128. Similar problem is other acronyms as HMW, LMW, CRM used across the manuscript and explained in accidental places. Mentioned problem and drawback of speciation analysis is poorly described and not addressed to applied methodology. In line 108 it is written “in my opinion” whereas there are two Authors. Whys this particular set of metals was chosen?

The aim of the study is unclear and discussion of chromatograms not convincing. In the results section Authors merely read what is already visible on the plots without deeper insight and meaning. The Discussion section does not add new insight and ends up without decisive conclusions.

Methodology is presented in unclear fashion. The accuracy and precision of applied approach is not discussed and enumerated observations are of un

The physicochemical properties of NADES are very sensitive to water concentration (even in trace amount) and are very hygroscopic. How authors checked influence of this factor on extraction process (and mechanism). The explanation in lines 415-416 is not sufficient.

Why composition listed in Table 1 are selected. Is is expectable that results of extractions are NADES concentration dependent?

In conclusion I want to state that I believe that paper might be publishable but definitely not in the present form. So I recommend major revision of the manuscript before resubmission.

Author Response

  • Although I do not feel competent for English language correction I feel that paper must be rewritten. The whole introduction is unclear, sentences are too complicated and vague. Please consider rewriting for increase clarity and enabling better comprehension.

The correction proof has been made before the submission of the manuscript.

  • The acronym SEC-ICP-MS is first time used in line 102 but explained in line 128. Similar problem is other acronyms as HMW, LMW, CRM used across the manuscript and explained in accidental places.

It was checked and rewritten.

  • Mentioned problem and drawback of speciation analysis is poorly described and not addressed to applied methodology.

The authors in the introduction section collect a description of the solvents used in the speciation analysis, indicating their advantages and disadvantages and the need to look for new solvents such as the presented NADES. The extraction process in the speciation analysis is a very important, difficult stage of the analytical procedure and the aim of the article was to present NADES as a new solution in the extraction in speciation analysis, therefore it seems to me that the description of other extractants used is important and just indicates the main purpose of the presented research.

  • In line 108 it is written “in my opinion” whereas there are two Authors. Whys this particular set of metals was chosen?

It was checked and rewritten.

  • The aim of the study is unclear and discussion of chromatograms not convincing. In the results section Authors merely read what is already visible on the plots without deeper insight and meaning. The Discussion section does not add new insight and ends up without decisive conclusions.

The aim of the presented investigation was to show the ability of used NADES extractants as solvents for the metal in speciation analysis. All data was obtained for natural material and CRM to increase the value of obtained results. We do hope that presented conclusions:

According to the results of our studies, the use of different NADES permits the extraction of various metals from a plant sample. Moreover, using other natural solvents of eutectic properties helps extract different species of a given metal. This possibility allows the planning of extraction of a sequence of metals and their compounds from plants. It enables speciation analysis of compounds bounded to insoluble parts of plants. Moreover, the application of sequential extraction with NADES permits effective extraction of metals to perform speciation analysis of biological and medical materials.

Give a new insight and shows the new possibility for preparation of sample in speciation analysis.

  • Methodology is presented in unclear fashion. The accuracy and precision of applied approach is not discussed and enumerated observations are of un

The reviewer's thought was unfinished - I cannot relate to this commentary. The information about the precision and accuracy have been added.

  • The physicochemical properties of NADES are very sensitive to water concentration (even in trace amount) and are very hygroscopic. How authors checked influence of this factor on extraction process (and mechanism). The explanation in lines 415-416 is not sufficient.

It was presented in our earlier studies (Osowska, N.; Ruzik, L. New Potentials in the Extraction of Trace Metal Using Natural Deep Eutectic Solvents (NADES). Food Anal. Methods 2019, doi:10.1007/s12161-018-01426-y.)

  • Why composition listed in Table 1 are selected. Is is expectable that results of extractions are NADES concentration dependent?

It was presented in our earlier studies (Osowska, N.; Ruzik, L. New Potentials in the Extraction of Trace Metal Using Natural Deep Eutectic Solvents (NADES). Food Anal. Methods 2019, doi:10.1007/s12161-018-01426-y.)

Round 2

Reviewer 3 Report

The revised version of the manuscript has been improved. It is now much more concise and might be interesting to readers of Molecules. I recommend publication as it is.